# Sparse Time-Frequency Distribution Reconstruction Using the Adaptive Compressed Sensed Area Optimized with the Multi-Objective Approach

**DOI:** 10.3390/s23084148

**Published:** 2023-04-20

**Authors:** Vedran Jurdana, Nikola Lopac, Miroslav Vrankic

**Affiliations:** 1Faculty of Engineering, University of Rijeka, 51000 Rijeka, Croatia; miroslav.vrankic@riteh.hr; 2Faculty of Maritime Studies, University of Rijeka, 51000 Rijeka, Croatia; 3Center for Artificial Intelligence and Cybersecurity, University of Rijeka, 51000 Rijeka, Croatia

**Keywords:** time-frequency distribution, sparse signal reconstruction, compressive sensing, Rényi entropy, multi-objective meta-heuristic optimization

## Abstract

Compressive sensing (CS) of the signal ambiguity function (AF) and enforcing the sparsity constraint on the resulting signal time-frequency distribution (TFD) has been shown to be an efficient method for time-frequency signal processing. This paper proposes a method for adaptive CS-AF area selection, which extracts the magnitude-significant AF samples through a clustering approach using the density-based spatial clustering algorithm. Moreover, an appropriate criterion for the performance of the method is formalized, i.e., component concentration and preservation, as well as interference suppression, are measured utilizing the information obtained from the short-term and the narrow-band Rényi entropies, while component connectivity is evaluated using the number of regions with continuously-connected samples. The CS-AF area selection and reconstruction algorithm parameters are optimized using an automatic multi-objective meta-heuristic optimization method, minimizing the here-proposed combination of measures as objective functions. Consistent improvement in CS-AF area selection and TFD reconstruction performance has been achieved without requiring a priori knowledge of the input signal for multiple reconstruction algorithms. This was demonstrated for both noisy synthetic and real-life signals.

## 1. Introduction

A time-frequency distribution (TFD) allows us to represent the signal energy jointly in time and frequency. Quadratic TFDs (QTFDs), the most commonly used TFDs in practice, introduce highly oscillatory artifacts, also called the cross-terms, for signals with multiple components or at least one non-linear frequency modulated (non-LFM) component [1]. The cross-terms are usually suppressed with the two-dimensional (2D) low-pass filters in the ambiguity function (AF) domain, but to the detriment of the useful components, called the auto-terms. The trade-off between the cross-term reduction and the auto-term resolution has led to a number of filtering methods over the years [1,2].

Compressive sensing (CS) has been a growing research topic [3,4,5] providing advances in various fields, including medicine [6], radar [7], and signal and image processing [8,9,10,11]. The CS-based methods allow signal reconstruction from a small subset of samples selected to favor a specific signal feature [12,13,14,15,16]. This allows a TFD reconstruction from a subset of signal samples from the AF [14,15,17,18]. It is desired that the selected CS-AF samples belong exclusively to the auto-terms located near the AF domain origin. Since the cross-terms are more oscillatory in the time-frequency (TF) domain, they are distributed over the rest of the AF domain. Therefore, the selection of the CS-AF samples is similar to the low-pass filtering of the AF. Such reduced CS-AF serves as input to the sparse reconstruction algorithm, which solves the unconstrained optimization problem, with the objective function emphasizing the TFD sparsity level [19,20].

The research on applying different optimization algorithms for CS area selection has been neglected, leaving the CS-AF area as a constant rectangular shape centered at the AF origin [12,14,15]. Such a CS-AF area usually contains unnecessary AF samples and does not follow the component trajectories in the AF. Moreover, it may contain noise samples or samples related to the interference caused by the non-LFM components, which may reappear in the resulting reconstructed TFD. In this paper, the parametrization of the CS-AF area proposed in [14] is introduced, which enhances its adaptability by allowing an arbitrary area shape. The separation of auto-terms and the cross-terms in the AF has been additionally ensured by the density-based spatial clustering of applications with noise (DBSCAN) algorithm. Among the various clustering methods, DBSCAN has been found to be the most suitable in terms of computational efficiency and the ability to form clusters without predicting the number of clusters [21,22,23]. The first goal of this paper is to improve the TFD reconstruction performance and demonstrate that the CS-AF area size can be substantially decreased by focusing solely on the most significant auto-term-specific AF samples.

The CS-based reconstruction leads to a decisive complexity in parameter optimization; its regularization parameter, set too large, leads to a discontinuity (oversparse) of the auto-term structure. This has led to primarily manual parameter selection [14,24], which is time-consuming and requires specialist knowledge of the used method and input signal nature. An optimization framework that uses a single global concentration measure where a lower or higher numerical value implies a better-performing TFD, thus not taking into account the lower or upper limit, usually favors a reconstructed TFD with discontinuous or completely absent auto-terms, making the commonly used global concentration measures unsuitable for solving this phenomenon.

This limitation was alleviated in [24] by coupling the concentration measure proposed in [25] with the one-dimensional (1D) localized Rényi entropy (LRE) information, namely, the short-term Rényi entropy (STRE) [26,27] and the narrow-band Rényi entropy (NBRE) [24,28]. The concentration measure was used to evaluate the component concentration and the suppression of cross-terms, while the mean squared error (MSE) between the local number of components (obtained from the STRE and NBRE) in starting and reconstructed TFD ensured that the auto-terms had been preserved [24]. The second goal of this paper is to demonstrate the effectiveness of the LRE method in measuring component concentration and cross-term suppression quality, thereby rendering the global concentration measure redundant as an objective function for optimizing CS-based reconstruction parameters. In addition to this, a new measure for the objective function is presented based on the number of regions with continuously connected samples, which can capture global component connectivity—a property that is not accounted for by either the localized approach of LRE or global concentration measures.

The proposed combination of objective functions has been implemented in a multi-objective meta-heuristic optimization algorithm for automatically optimizing the CS-based reconstruction parameters without any a priori signal information. The proposed CS-AF area selection, the objective functions, and the TFD reconstruction performance were tested on synthetic and real-life signals with additive noise using the Rényi entropy-based shrinkage (RTwIST) reconstruction algorithm proposed in [24]. The proposed CS-AF area selection procedure has been verified on several state-of-the-art reconstruction algorithms, demonstrating its robustness and effectiveness.

In a broader context, an improvement in the accuracy and efficiency of CS-based methods due to the proposed CS-AF area selection makes them more suitable for a variety of applications, such as signal identification, classification, and content extraction in noisy environments. Additionally, the proposed objective functions for automatic optimization streamline the optimization process, eliminating the need for user input other than the signal itself. This feature makes the optimization process more user-friendly and accessible to individuals without expertise in CS-based methods, as it enables them to obtain meaningful results.

The rest of this paper is organized as follows. The proposed method is described in Section 2. The obtained results are presented and discussed in Section 3. Finally, the paper’s conclusions are summarized in Section 4.

## 2. Materials and Methods

### 2.1. Time-Frequency Signal Analysis

Let us consider a multi-component non-stationary signal z(t), an analytic associate of a real signal s(t), expressed as:(1)$$z\left(t\right)=\sum_{i=1}^{NC}a_{i}\left(t\right)e^{j\varphi_{i}\left(t\right)},$$
where NC is the number of components, while $a_{i}\left(t\right)$ and $\varphi_{i}\left(t\right)$ are the instantaneous amplitude and instantaneous phase of the signal’s *i*-th component, respectively. Ideal TFD $\hat{\rho }\left(t,f\right)$ at any given time is a unit delta function at the instantaneous frequency (IF) $f_{0_{i}}\left(t\right)$ of the *i*-th component: (2)$$\hat{\rho }\left(t,f\right)=\sum_{i=1}^{NC}a_{i}^{2}\left(t\right)\delta \left(f-f_{0_{i}}\left(t\right)\right),$$
(3)$$f_{0_{i}}\left(t\right)=\frac{1}{2\pi }\frac{d\varphi_{i}\left(t\right)}{dt}.$$

In most practical examples, the ideal TFD is not achievable since practical TFDs are not perfectly localized and can suffer from the cross-terms [1].

The Wigner–Ville Distribution (WVD), the most commonly used TFD, is defined as [1]: (4)$$W_{z}\left(t,f\right)=\int_{-\infty }^{\infty }z\left(t+\frac{\tau }{2}\right)z^{*}\left(t-\frac{\tau }{2}\right)e^{-j2\pi f\tau }d\tau ,$$
and it gives an almost perfect IF estimate for a signal with a single LFM component in the TF plane. However, the proneness to the cross-terms (when dealing with the multi-component signals) has created a need for adequate cross-term suppression. In the AF $A_{z}\left(\nu ,\tau \right)$, calculated as: (5)$$A_{z}\left(\nu ,\tau \right)=\int_{-\infty }^{\infty }\int_{-\infty }^{\infty }W_{z}\left(t,f\right)e^{j2\pi (f\tau -\nu t)}dtdf,$$
the highly oscillatory cross-terms can be suppressed with a 2D low-pass filter, defining a QTFD class of TFDs, ρ(t,f): (6)$$A_{z}\left(\nu ,\tau \right)=A_{z}\left(\nu ,\tau \right)g\left(\nu ,\tau \right),$$
(7)$$\rho \left(t,f\right)=\int_{-\infty }^{\infty }\int_{-\infty }^{\infty }A_{z}\left(\nu ,\tau \right)e^{j2\pi (\nu t-f\tau )}d\nu d\tau ,$$
where $g\left(\nu ,\tau \right)$ is the low-pass filter kernel in the AF. The kernel design usually involves a trade-off between the auto-term concentration and the cross-term suppression [1].

### 2.2. The Localized Rényi Entropy

A global measure for signal complexity in the TF plane, known as the Rényi entropy R(ρ(t,f)), is defined as [29,30,31]: (8)$$R\left(\rho \left(t,f\right)\right)=\frac{1}{1-\alpha_{R}}\log_{2}\int_{-\infty }^{\infty }\int_{-\infty }^{\infty }{\left(\frac{\rho (t,f)}{\int_{-\infty }^{\infty }\int_{-\infty }^{\infty }\rho \left(t,f\right)\;dtdf}\right)}^{\alpha_{R}}dtdf,$$
where for the odd integer parameter $\alpha_{R}>2$, the cross-terms get integrated-out from the QTFD ρ(t,f), which is normalized with respect to its total energy [27,29].

Using the counting property of the Rényi entropy, this global approach has been modified to extract the local number of signal components per time slice $t_{0}$ using the STRE [27], calculated as: (9)$$NC_{t}^{\rho (t,f)}\left(t_{0}\right)=2^{R\left(\Delta_{t_{0}}\left\{\rho \left(t,f\right)\right\}\right)-R\left(\Delta_{t_{0}}\left\{\rho_{\mathrm{ref}}\left(t,f\right)\right\}\right)},$$
where $t_{0}$ is the observed time slice, while $\rho \left(t,f\right)$ and $\rho_{\mathrm{ref}}\left(t,f\right)$ are the considered and reference QTFD, respectively. The time-localization operator $\Delta_{t_{0}}$ sets all TFD samples to zero, except those in the vicinity of $t_{0}$: (10)$$\Delta_{t_{0}}\left\{\rho \left(t,f\right)\right\}=\left\{\begin{array}{cc} \rho (t,f), & t_{0}-\Theta_{t}<t<t_{0}+\Theta_{t},\\ 0, & \mathrm{otherwise}, \end{array}\right.$$
where $\Theta_{t}$ is the parameter controlling the time window length. The reference signal is chosen to be a cosine signal with an amplitude of 1 and a constant normalized frequency of 0.1, providing a reference energy of a single component in every time slice [27].

Further analysis has been done in [24,28], revealing the shortcomings of the STRE for certain signal types and introducing the NBRE to tackle them. Using the NBRE, one can calculate the number of local components per frequency slice $f_{0}$ by replacing the time-localization parameter in Equation (Equation 9) with the frequency-localization operator: (11)$$\Delta_{f_{0}}\left\{\rho \left(t,f\right)\right\}=\left\{\begin{array}{cc} \rho (t,f), & f_{0}-\Theta_{f}<f<f_{0}+\Theta_{f},\\ 0, & \mathrm{otherwise}, \end{array}\right.$$
where $f_{0}$ is the observed frequency slice and $\Theta_{f}$ is the frequency window length. The reference signal is a delta function centered at t=15 [24]. The optimal time and frequency window lengths should capture the signal’s relevant time-frequency structure. A window length that is too large may fail to capture the fast-changing instantaneous frequency of the signal’s auto-terms, whereas a window length that is too small may pronounce noise or cross-term energy regions, and misclassify them as auto-terms, particularly when the TFD is not perfectly free of interference. Recommended ranges for window length are available in [26,32,33]. Furthermore, $\rho \left(t,f\right)$ and $\rho_{\mathrm{ref}}\left(t,f\right)$ have to be obtained with the same AF kernel in order for the comparison to be valid. In this work, a high-performance separable kernel-based QTFD has been used, namely, the extended modified B distribution (EMBD), whose kernel is defined as: (12)$$g\left(\nu ,\tau \right)=\frac{\int_{-\infty }^{\infty }{\cos h}^{-2\beta_{E}}\left(t\right)e^{j2\pi \nu t}dt}{\int_{-\infty }^{\infty }{\cos h}^{-2\beta_{E}}\left(t\right)dt}{\cos h}^{-2\alpha_{E}}\left(\tau \right),$$
where $0\leq \alpha_{E}\leq 1$ and $0\leq \beta_{E}\leq 1$ are the frequency and time smoothing parameters, respectively [1,14,34].

### 2.3. Sparse Time-Frequency Distributions

Enhancement of the TFD resolution can be achieved by using the compressive sensing approach. Namely, a set of samples is taken from the AF, denoted as CS-AF $A_{z}^{\prime }\left(\nu ,\tau \right)$:(13)$$A_{z}^{\prime }\left(\nu ,\tau \right)=\left\{\begin{array}{cc} A_{z}\left(\nu ,\tau \right), & \left(\nu ,\tau \right)\in C_{\Omega },\\ 0, & \mathrm{otherwise}, \end{array}\right.$$
where $A_{z}\left(\nu ,\tau \right)$ is the matrix representation of the AF and $C_{\Omega }$ is the index set of CS-AF samples. Note that our goal is to reconstruct a TFD without the cross-terms, deviating from the standard CS goal, which aims to exactly reconstruct the starting signal. Hence, the indices $C_{\Omega }$ are not chosen randomly; they are purposely focused on the TFD auto-terms. The connection between the sparse TFD and the CS-AF is: (14)$$\vartheta_{z}\left(t,f\right)=\psi^{H}\cdot A_{z}^{\prime }\left(\nu ,\tau \right),$$
where ψ is the domain transformation matrix representing the 2D Fourier transform equivalent to Equation (Equation 5), while ${\left(\cdot \right)}^{H}$ is the Hermitian transpose. Since $\mathrm{card}\left(A_{z}^{\prime }\left(\nu ,\tau \right)\right)<<\mathrm{card}\left(\vartheta_{z}\left(t,f\right)\right)$, there are multiple solutions of Equation (Equation 14), and the goal of the sparse TFD reconstruction is to find an optimal solution by minimizing the enforced sparsity-inducing function. One of the commonly used sparsity-inducing functions is the $\ell_{0}$-norm, which counts the number of non-zero elements, formalizing the following unconstrained optimization problem [14,19,35]: (15)$$\begin{array}{c} \vartheta_{z}^{\ell_{0}}\left(t,f\right)=\arg\min_{\vartheta_{z}\left(t,f\right)}\left|\right|\vartheta_{z}\left(t,f\right){\left|\right|}_{0},\\ \;\mathrm{subject}\;\mathrm{to}:\;||\vartheta_{z}\left(t,f\right)-\psi^{H}A_{z}^{\prime }\left(\nu ,\tau \right){\left|\right|}_{2}^{2}\leq \epsilon , \end{array}$$
where ϵ is the solution tolerance. The proximity operator has been introduced to Equation (Equation 15) in order to rewrite this problem in a closed-form expression [35]: (16)$$\vartheta_{z}^{\ell_{0}}\left(t,f\right)={\mathrm{hard}}_{\sqrt{2\lambda }}\left\{\vartheta_{z}\left(t,f\right)\right\},$$
where ${\mathrm{hard}}_{\sqrt{2\lambda }}\left\{\vartheta_{z}\left(t,f\right)\right\}$ is a hard-thresholding function defined as: (17)$${\mathrm{hard}}_{\sqrt{2\lambda }}\left\{\vartheta_{z}\left(t,f\right)\right\}=\left\{\begin{array}{cc} \vartheta_{z}\left(t,f\right), & \vartheta_{z}\left(t,f\right)\geq \sqrt{2\lambda },\\ 0, & \mathrm{otherwise}, \end{array}\right.$$
and λ>0 is the regularization parameter introducing the mentioned complexity in its optimal value selection.

### 2.4. The Rényi Entropy-Based TFD Reconstruction Algorithm

The RTwiST algorithm presented in [24,28] is based on the two-step iterative shrinkage/thresholding (TwIST) algorithm [20] given as: (18)$${\left[\vartheta_{z}^{\ell_{0}}\left(t,f\right)\right]}^{\left[n+1\right]}=\left(1-\alpha \right){\left[\vartheta_{z}^{\ell_{0}}\left(t,f\right)\right]}^{\left[n-1\right]}+\left(\alpha -\beta \right){\left[\vartheta_{z}^{\ell_{0}}\left(t,f\right)\right]}^{\left[n\right]}+\beta {\left[\zeta_{z}\left(t,f\right)\right]}^{\left[n+1\right]},$$
(19)$${\left[\zeta_{z}\left(t,f\right)\right]}^{\left[n+1\right]}={\mathrm{hard}}_{\sqrt{2\lambda }}\left\{{\left[\zeta_{z}^{\prime }\left(t,f\right)\right]}^{\left[n+1\right]}\right\},$$
(20)$${\left[\zeta_{z}^{\prime }\left(t,f\right)\right]}^{\left[n+1\right]}={\left[\vartheta_{z}^{\ell_{0}}\left(t,f\right)\right]}^{\left[n\right]}+\psi^{H}\left(A_{z}^{\prime }\left(\nu ,\tau \right)-\psi {\left[\vartheta_{z}^{\ell_{0}}\left(t,f\right)\right]}^{\left[n\right]}\right),$$
where 0≤α≤1 and 0≤β≤2α are the TwIST relaxation parameters. The RTwIST algorithm replaces the hard-thresholding operator with the Rényi entropy-based shrinking operator, denoted as ${\mathrm{shrink}}_{t,f}\left\{{\left[\zeta_{z}^{\prime }\left(t,f\right)\right]}^{\left[n+1\right]}\right\}$. The shrinkage operator operates on the individual time and frequency slices, utilizing the number of local components, $NC_{t}$ and $NC_{f}$, respectively. More precisely, every slice’s local maximum is associated with an area calculated as a sum of samples between the local minima to the left and to the right of the observed maximum. All samples not belonging to the $NC_{t}$ (or $NC_{f}$) largest areas are set to zero. In order to further enhance the TFD resolution, two threshold parameters have been introduced: $0\leq \delta_{t},\delta_{f}\leq 1$, which further narrows the auto-term related regions. It has been shown that setting $0.88\leq \delta_{t},\delta_{f}\leq 0.94$ provides satisfying results [24]. The enhanced adaptability of the here-proposed CS-AF area selection method, combined with the proposed optimization framework, revealed some well-performing parameter combinations where $\delta_{t}$ and $\delta_{f}$ are outside this range.

In the next step, the two TFDs obtained from the time and frequency-based shrinkage, denoted as $\zeta_{z}^{t}\left(t,f\right)$ and $\zeta_{z}^{f}\left(t,f\right)$, respectively, are averaged: (21)$${\left[\zeta_{z}\left(t,f\right)\right]}^{\left[n+1\right]}=p\cdot {\left[\zeta_{z}^{t}\left(t,f\right)\right]}^{\left[n+1\right]}+\left(1-p\right)\cdot {\left[\zeta_{z}^{f}\left(t,f\right)\right]}^{\left[n+1\right]},$$
where 0≤p≤1 is the weighting parameter emphasizing the respective TFD, depending if the components’ slopes are more aligned towards the time or frequency axis [24]. More precisely, p>0.5 if the signal components are more aligned with the time axis, while p<0.5 for signal components more aligned with the frequency axis [24]. The RTwIST algorithm is defined by replacing Equation (Equation 19) with Equation (Equation 21), and the solution is obtained by iterating them until the stopping criterion ϵ is satisfied or the maximum number of iterations $N_{it}$ is reached. In [24], it was shown that RTwIST outperforms state-of-the-art reconstruction algorithms that use a hard-thresholding operator with the parameter λ. However, despite the absence of the parameter λ, setting the parameters $\delta_{t}$ and $\delta_{f}$ too large or emphasizing the respective TFD with inconsistent components (parameter *p*) also leads to an oversparse reconstructed TFD.

### 2.5. The Proposed Adaptive CS-AF Area Selection

#### 2.5.1. Ambiguity Function Thresholding

The result of the sparse TFD reconstruction is highly dependent on the proper selection of the CS-AF area. It has been shown that too few samples reduce TF resolution and fail to reconstruct higher-polynomial FM signal components with low amplitude. On the other hand, using too many samples leads to the unwanted reconstruction of the cross-terms [14]. Furthermore, this problem is further emphasized by the RTwIST algorithm, as it can decrease the classification accuracy of the shrinkage operator, i.e., the cross-term-related samples can be included as one of the $NC_{t}\left(t\right)$ (or $NC_{f}\left(f\right)$) components, while discarding the auto-term-related samples.

Since the goal of sparse reconstruction is to obtain a cross-terms-free TFD, the authors in [14] have proposed an adaptive $N_{\tau }^{\prime }\times N_{\nu }^{\prime }$ rectangular CS-AF area, where $N_{\tau }^{\prime }$ and $N_{\nu }^{\prime }$ denote the numbers of lag and Doppler bins, respectively. This area is centered around the AF origin, while its vertices are positioned adjacent to the first pair of cross-terms, capturing the cross-terms-free area around the AF origin as large as possible. Increasing the cardinality of this area compared to the static $\sqrt{N_{t}}\times \sqrt{N_{t}}$ area ($N_{t}$ being the number of time samples) lowers the requirements of the reconstruction algorithm. However, the complete $N_{\tau }^{\prime }\times N_{\nu }^{\prime }$ area has two limitations; first, it emphasizes the linear part behavior of the auto-terms, which can deteriorate the trajectories of higher polynomial FM components. Second, its strict rectangular shape may include AF samples unrelated to the auto-terms as $N_{\tau }$ and $N_{\nu }$ increase, which reduces the overall reconstruction performance.

In this paper, a parametrization of the above-mentioned method is proposed motivated by an assumption that the reconstruction performance can be improved by selecting only the most significant AF samples within the original CS-AF area, denoted as $A_{z_{\Gamma }}\left(\nu ,\tau \right)$, with the optional inclusion of the external AF samples from the rest of the AF area, denoted as $A_{z_{\Upsilon }}\left(\nu ,\tau \right)$. Thus, the following database matrix Ω, which will serve as an input of the DBSCAN algorithm, is constructed as follows: (22)$$\Omega =\left\{\begin{array}{cc} A_{z}\left(\nu ,\tau \right), & |A_{z_{\Gamma }}\left(\nu ,\tau \right)\left|\geq \Gamma \cup \right|A_{z_{\Upsilon }}\left(\nu ,\tau \right)|\geq \Upsilon ,\\ 0, & \mathrm{otherwise}, \end{array}\right.$$
where Γ and Υ parameters are the minimum normalized magnitude of the AF sample selected from $A_{z_{\Gamma }}\left(\nu ,\tau \right)$ and $A_{z_{\Upsilon }}\left(\nu ,\tau \right)$, respectively. It is important to note that the auto-terms and the cross-terms have the following AF properties [36,37]:The auto-terms are concentrated along trajectories passing through the AF origin. These trajectories follow the components’ IF law.The auto-term magnitude peaks at the AF origin and steadily decreases.The auto-terms and cross-terms may intersect in the AF.

The majority of the samples within the $A_{z_{\Gamma }}\left(\nu ,\tau \right)$ are important in the TFD reconstruction since they represent signal auto-terms. However, it is possible to lower the number of samples by excluding the low-magnitude ones which are often contained within the $A_{z_{\Gamma }}\left(\nu ,\tau \right)$ due to its strict rectangular shape. These samples may contain noise or be cross-term-related due to the non-linear auto-term behavior, and, as such, would be reconstructed in the resulting TFD. Because of this, using a relatively low Γ value is recommended, as $A_{z_{\Gamma }}\left(\nu ,\tau \right)$ could also contain the auto-terms with low magnitude.

On the other hand, in $A_{z_{\Upsilon }}\left(\nu ,\tau \right)$, the goal is to expand the CS-AF area with the auto-term-related samples in order to provide more information about the non-linear auto-term behavior. Due to the above-mentioned second property, the cross-terms in $A_{z_{\Upsilon }}\left(\nu ,\tau \right)$ have a magnitude that can surpass the magnitude of the auto-terms. Because of this, the low Γ value used in $A_{z_{\Gamma }}\left(\nu ,\tau \right)$ is inadequate. Thus, in order to guarantee the exclusive inclusion of the auto-term-related samples, it is recommended to use a much higher Υ value in $A_{z_{\Upsilon }}\left(\nu ,\tau \right)$. It is worth noting that the optimal values of Γ and Υ are signal-dependent and lie within the range of [0,1]. Consequently, these parameters will be optimized together with the reconstruction parameters for each signal example to determine their optimal values. Furthermore, for (Γ,Υ)=(0,1), the CS-AF is equivalent to the original $N_{\tau }^{\prime }\times N_{\nu }^{\prime }$ area. The idea behind this is not to discriminate between the auto-term and the cross-term samples but rather to lower the card(Ω) while preserving as much of the auto-terms as possible in order to decrease the computational complexity of the clustering algorithm. In the next step, the DBSCAN algorithm will be utilized to make the final classification decision.

#### 2.5.2. The Density-Based Spatial Clustering

The next and final step involves extracting a single cluster from Ω centered at the AF origin, denoted as $C_{\Omega }$ in Equation (Equation 13), satisfying the previously outlined auto-term properties. In order to do that, the obtained Ω serves as the input for a data clustering problem. Given that it requires the extraction of only a single cluster with the known position in the AF (defined by the auto-term properties), the DBSCAN has been used since it does not require a predefined cluster number. Furthermore, the DBSCAN has strong flexibility in the cluster shapes and sizes, which is essential for signals with various auto-term trajectories. These advantages, along with the lower computational complexity and the ability to exclude outliers and noise-related samples from a cluster, highlight the DBSCAN over the partitioning and hierarchical methods [22,23,38] for our use case.

The DBSCAN aims to detect clusters in Ω that satisfy the minimum density criterion and are separated by an empty area or an area with lower sample density [21,39,40,41,42]. The DBSCAN performance is controlled with two parameters. The first parameter is the radius ε defining the ε-neighborhood of the AF sample $A_{z}\left(\nu_{p},\tau_{p}\right)$: (23)$$N_{\varepsilon }\left(A_{z}\left(\nu_{p},\tau_{p}\right)\right)=\left\{A_{z}\left(\nu_{q},\tau_{q}\right)\in \Omega \;|\;d\left(A_{z}\left(\nu_{p},\tau_{p}\right),A_{z}\left(\nu_{q},\tau_{q}\right)\right)\leq \varepsilon \right\},$$
where $A_{z}\left(\nu_{q},\tau_{q}\right)$ denotes the AF sample in the ε-neighborhood of the $A_{z}\left(\nu_{p},\tau_{p}\right)$, within Euclidean distance: (24)$$d\left(A_{z}\left(\nu_{p},\tau_{p}\right),A_{z}\left(\nu_{q},\tau_{q}\right)\right)=\sqrt{{\left(\nu_{p}-\nu_{q}\right)}^{2}+{\left(\tau_{p}-\tau_{q}\right)}^{2}}.$$

The second DBSCAN parameter, minPts, controls the minimum number of samples within the ε-neighborhood of the observed AF sample.

For our clustering problem, the parameter values have been set to minPts=4 and ε=3, according to recommendations given in [21,43]. Furthermore, border samples have been merged with core samples in order to form an even cluster, which is more consistent with the concepts of a density level set [21,44]. The computational efficiency of DBSCAN is $O(n^{2})$ or O(n·logn) for special data structures. In order to enhance the computational efficiency, DBSCAN does not perform density estimation in between points. Instead, it transitively clusters all samples within the ε-neighborhood of the core sample [21].

### 2.6. The Proposed Objective Functions for the Multi-Objective Optimization Method

#### 2.6.1. The Measure Based on the Localized Rényi Entropy

The purpose of the objective functions is to guide the optimization process towards a high-performing reconstructed TFD accurately, i.e., a TFD with high auto-term resolution and preservation, without reconstructing cross-term or noise-related TF samples. As mentioned above, the CS-based reconstruction method involves optimization difficulty as its parameters may cause the loss of auto-terms; therefore, the use of measures without lower or upper limit and position information is excluded. Given that the number of components can vary locally, the loss of components can occur anywhere in a TFD. For this reason, the local approach and extraction of the local number of components in the starting TFD (TFD with fully preserved auto-terms) provide valuable information for the optimization process [24]. More precisely, the objective functions are formalized as two mean squared errors (MSEs) between the local number of components in the starting, ρ(t,f), and reconstructed TFD, $\vartheta_{z}^{\ell_{0}}\left(t,f\right)$, obtained from the STRE (notation *t*) and NBRE (notation *f*), as: (25)$${\mathrm{MSE}}_{t}=\frac{1}{N_{t}}\sum_{t=1}^{N_{t}}{\left(\frac{NC_{t}^{\rho (t,f)}\left(t\right)-NC_{t}^{\vartheta_{z}^{\ell_{0}}\left(t,f\right)}\left(t\right)}{\max\left(NC_{t}^{\rho (t,f)}\left(t\right),NC_{t}^{\vartheta_{z}^{\ell_{0}}\left(t,f\right)}\left(t\right)\right)}\right)}^{2},$$
(26)$${\mathrm{MSE}}_{f}=\frac{1}{N_{f}}\sum_{f=1}^{N_{f}}{\left(\frac{NC_{f}^{\rho (t,f)}\left(f\right)-NC_{f}^{\vartheta_{z}^{\ell_{0}}\left(t,f\right)}\left(f\right)}{\max\left(NC_{f}^{\rho (t,f)}\left(f\right),NC_{f}^{\vartheta_{z}^{\ell_{0}}\left(t,f\right)}\left(f\right)\right)}\right)}^{2},$$
where $N_{f}$ is the number of frequency bins. In [24], the authors have combined this measure with the global concentration measure proposed in [25], given as: (27)$$M_{z}^{S}=\frac{1}{N_{t}N_{f}}{\left[\sum_{t=1}^{N_{t}}\sum_{f=1}^{N_{f}}{\left|\vartheta_{z}^{\ell_{0}}\left(t,f\right)\right|}^{1/2}\right]}^{2},$$
where the lower $M_{z}^{S}$ value indicated a reconstructed TFD with higher auto-term concentration and cross-term suppression, while the lower ${\mathrm{MSE}}_{t,f}={\mathrm{MSE}}_{t}+{\mathrm{MSE}}_{f}$ value indicated better auto-term preservation.

However, simulations on various test signals have shown that using only the ${\mathrm{MSE}}_{t}$ and ${\mathrm{MSE}}_{f}$ as objective functions with the ideal TFDs of the reference signals for $NC_{t}^{\vartheta_{z}^{\ell_{0}}\left(t,f\right)}\left(t\right)$ and $NC_{f}^{\vartheta_{z}^{\ell_{0}}\left(t,f\right)}\left(f\right)$ when considering the STRE and NBRE, respectively, ensures a high-performing reconstructed TFD. In this work, our goal is to show that a reconstructed TFD with low resolution or component inconsistency leads to considerable differences between the respective Rényi entropies, resulting in an incorrect estimate of the local number of components and, ultimately, misclassification with respect to the ${\mathrm{MSE}}_{t}$ and ${\mathrm{MSE}}_{f}$ values. This discovery precludes the need for the $M_{z}^{S}$ measure and reduces the computational complexity of the optimization process, which requires multiple executions of objective functions.

#### 2.6.2. The Proposed Number of Regions with Continuously Connected Samples

Since ${\mathrm{MSE}}_{t}$ and ${\mathrm{MSE}}_{f}$ operate on isolated time or frequency slices, the position and connectivity of signal components through different slices is not taken into account. Hence, as the third objective function, minimizing the number of regions with continuously connected TFD samples, denoted with $N_{r}$, is proposed. In this work, each sample with location (x,y) within one region is connected to at least one sample in its eight-neighborhood set. With that criterion, it is desired that the resulting reconstructed TFD obtained by the optimization have its components as close as possible to continuous IF trajectories without any interruption.

Thus, this objective evaluates the component consistency, where a lower $N_{r}$ value indicates that the components, i.e., the auto-terms, have higher consistency. Since the presence of interference and noise in the reconstructed TFD involves additional regions, a lower $N_{r}$ value can also indicate reconstructed TFD with fewer interference and noise samples. Ideally, $N_{r}$ would equal the global number of signal components in the TF plane; however, in practice, $N_{r}$ is usually larger since reconstructed TFD components are not perfectly connected in individual IF trajectories. Thus, the $N_{r}$ value is not interpreted as a measure of the global number of components but rather as a measure supplementing the ${\mathrm{MSE}}_{t}$ and ${\mathrm{MSE}}_{f}$ values.

### 2.7. The Multi-Objective Meta-Heuristic Optimization

The optimization scheme proposed in [45] constructs an optimized TFD by selecting the optimal TFD for each time instant from a set of arbitrarily generated TFDs with the minimum local number of components and the smallest entropy. In order to obtain the optimal TFD, the optimization scheme implies at least one cross-terms-free TFD for each time instant within the chosen set of TFDs. In the context of sparse reconstruction, the arbitrarily chosen parameters of the CS-AF area or the reconstruction algorithm could lead to the absence of essential components in a certain time instant of the reconstructed TFD. Therefore, the optimization procedure in [45] would incorrectly classify a reconstructed TFD with inconsistent components as the best one. Moreover, given the number of parameters involved in our context, it would be inconvenient and time- and memory-consuming to generate a large set of reconstructed TFDs.

Classical gradient-based optimization techniques, such as the gradient descent method (GDM), are highly dependent on the choice of initial parameter values and therefore require the user to have knowledge of the TFD and the analyzed signal [46]. Furthermore, they generate a single point at each iteration which might lead to many iterations, and reaching a global minimum/maximum is not guaranteed for a non-convex function. Finally, GDM is not derivative-free. The classical Nelder–Mead optimization algorithm is derivative-free, but its performance depends on the correct choice of initial parameters. The disadvantages of these methods have been addressed in [46] by using a hybrid genetic algorithm that provides an automatic approach to optimize QTFDs while minimizing a single objective concentration measure. In our context, all the mentioned optimization techniques using a single objective suffer from the same problem: minimizing a single concentration measure leads to an oversparse (or completely empty) reconstructed TFD.

As a solution, the use of multiple here-defined objective functions is imposed to formalize a multi-objective optimization problem for the RTwIST algorithm as: (28)$$\begin{array}{c} \min\{{\mathrm{MSE}}_{t},{\mathrm{MSE}}_{f},N_{r}\left(\alpha ,\beta ,p,\delta_{t},\delta_{f},\Gamma ,\Upsilon \right)\},\\ s.t.\;\;\alpha ,p,\delta_{t},\delta_{f},\Gamma ,\Upsilon \in \left[0,1\right],\beta \in \left[0,2\alpha \right]. \end{array}$$

In this work, the multi-objective optimization method based on particle swarm optimization (MOPSO) [47,48,49] has been used. In multi-objective optimization, the feasible solution does not minimize all objective functions simultaneously, i.e., one objective cannot be improved without degrading at least one of the other objectives. Therefore, the multi-objective algorithm constructs all non-dominated solutions in the Pareto front [47,48]. The dominance relationship is a fundamental concept in multi-objective optimization that identifies superior solutions among a set of alternatives. Specifically, a solution x is said to dominate another solution y if x is at least as good as y in all objectives, namely, ${\mathrm{MSE}}_{t}\left(x\right)\leq {\mathrm{MSE}}_{t}\left(y\right)$, ${\mathrm{MSE}}_{f}\left(x\right)\leq {\mathrm{MSE}}_{f}\left(y\right)$, $N_{r}\left(x\right)\leq N_{r}\left(y\right)$, and strictly better than y in at least one objective, that is, either ${\mathrm{MSE}}_{t}\left(x\right)<{\mathrm{MSE}}_{t}\left(y\right)$, ${\mathrm{MSE}}_{f}\left(x\right)<{\mathrm{MSE}}_{f}\left(y\right)$, or $N_{r}\left(x\right)<N_{r}\left(y\right)$. The Pareto front characterizes the set of non-dominated solutions which form the trade-off boundary between competing objectives. The optimal solution among the Pareto front has been chosen by the fuzzy satisfying method (FSM) [50] as used in [24].

## 3. Results and Discussion

The here-proposed algorithm performance has been tested on a synthetic signal $z_{\mathrm{SS}}$, with $N_{t}=256$ samples, composed of two parabolic FM components embedded in additive white Gaussian noise (AWGN) with SNR=5 dB, as well as on the real-life gravitational-wave signal (This research has made use of data, software, and/or web tools obtained from the LIGO Open Science Center (https://losc.ligo.org (accessed on 1 February 2023)), a service of LIGO Laboratory and the LIGO Scientific Collaboration. LIGO is funded by the U.S. National Science Foundation.) $z_{\mathrm{RS}}$ [51,52], with $N_{t}=216$ samples. The WVDs of the considered signals and their respective AFs with the $N_{\tau }^{\prime }\times N_{\nu }^{\prime }$ areas selected by the method in [14] are shown in Figure 1.

The performance of the proposed CS-AF area selection was tested using the following state-of-the-art reconstruction algorithms: the RTwIST [24], the TwIST [20], the Sparse reconstruction by separable approximation (SpaRSA) [53], and the Split augmented Lagrangian shrinkage algorithm (SALSA) [54]. Note that the input parameters of all evaluated reconstruction algorithms (RTwIST, TwIST, SpaRSA, SALSA) have been optimized using the same procedure described in Section 2.6.

For the MOPSO, the numbers of iterations and swarm size have been set to 100 and the maximum number of particles in the Pareto front has been set to 50 as in [24], while the inertia weight and the coefficients controlling the stochastic terms have been set according to [48]. The localized Rényi entropy has been calculated using the parameter $\alpha_{R}=3$ and $\Theta_{t}=\Theta_{f}=11$ as used in [14,24,28], while the EMBD parameters $\alpha_{E}=0.09,\beta_{E}=0.18$ and $\alpha_{E}=0.12,\beta_{E}=0.13$ for the signals $z_{\mathrm{SS}}$ and $z_{\mathrm{RS}}$, respectively, have been optimized using the measure proposed in [25]. Finally, the parameters $\epsilon =10^{-3}$ and $N_{it}=100$ have been set as in [14,24,34].

In order to further quantify and compare the performance of the proposed CS-AF area construction, additional performance indicators have been used; the mean absolute error (MAE) and the maximum absolute error (MAX) between the local number of components in starting and reconstructed TFDs: (29)$${\mathrm{MAE}}_{t}=\frac{1}{N_{t}}\sum_{t=1}^{N_{t}}|\frac{NC_{t}^{\rho (t,f)}\left(t\right)-NC_{t}^{\vartheta_{z}^{\ell_{0}}\left(t,f\right)}\left(t\right)}{\max\left(NC_{t}^{\rho (t,f)}\left(t\right),NC_{t}^{\vartheta_{z}^{\ell_{0}}\left(t,f\right)}\left(t\right)\right)}|,$$
(30)$${\mathrm{MAE}}_{f}=\frac{1}{N_{f}}\sum_{f=1}^{N_{f}}|\frac{NC_{f}^{\rho (t,f)}\left(f\right)-NC_{f}^{\vartheta_{z}^{\ell_{0}}\left(t,f\right)}\left(f\right)}{\max\left(NC_{f}^{\rho (t,f)}\left(f\right),NC_{f}^{\vartheta_{z}^{\ell_{0}}\left(t,f\right)}\left(f\right)\right)}|,$$
(31)$${\mathrm{MAX}}_{t}=\max_{t=1,\ldots ,N_{t}}\left(|\frac{NC_{t}^{\rho (t,f)}\left(t\right)-NC_{t}^{\vartheta_{z}^{\ell_{0}}\left(t,f\right)}\left(t\right)}{\max\left(NC_{t}^{\rho (t,f)}\left(t\right),NC_{t}^{\vartheta_{z}^{\ell_{0}}\left(t,f\right)}\left(t\right)\right)}|\right),$$
(32)$${\mathrm{MAX}}_{f}=\max_{f=1,\ldots ,N_{f}}\left(|\frac{NC_{f}^{\rho (t,f)}\left(f\right)-NC_{f}^{\vartheta_{z}^{\ell_{0}}\left(t,f\right)}\left(f\right)}{\max\left(NC_{f}^{\rho (t,f)}\left(f\right),NC_{f}^{\vartheta_{z}^{\ell_{0}}\left(t,f\right)}\left(f\right)\right)}|\right).$$

The proposed objective functions and performance indicators have been compared with commonly used concentration measures: the global Rényi entropy, *R*, given in Equation (Equation 8), the concentration measure, $M_{z}^{S}$, given in Equation (Equation 27), the ratio of norm-based measure [1] defined as: (33)$$RN=\frac{\sum_{t=1}^{N_{t}}\sum_{f=1}^{N_{f}}{\left(\vartheta_{z}^{\ell_{0}}\left(t,f\right)\right)}^{4}}{{\left(\sum_{t=1}^{N_{t}}\sum_{f=1}^{N_{f}}{\left(\vartheta_{z}^{\ell_{0}}\left(t,f\right)\right)}^{2}\right)}^{2}},$$
and the Hoyer measure defined as [55]: (34)$$HM=\left(\sqrt{H}-\frac{\sum_{t=1}^{N_{t}}\sum_{f=1}^{N_{f}}\left|\vartheta_{z}^{\ell_{0}}\left(t,f\right)\right|}{\sqrt{\sum_{t=1}^{N_{t}}\sum_{f=1}^{N_{f}}{\left|\vartheta_{z}^{\ell_{0}}\left(t,f\right)\right|}^{2}}}\right){\left(\sqrt{H}-1\right)}^{-1},$$
where *H* is the total size of the TFD: $H=N_{t}\times N_{f}$. For RN and HM, a larger value represents a more concentrated and desirable TFD, while the opposite is true for the measures *R* and $M_{z}^{S}$.

### 3.1. CS-AF Area Selection

Figure 2 shows the clustered CS-AF areas. By using the proposed method, the $C_{\Omega }$ can be reduced from the $N_{\tau }^{\prime }\times N_{\nu }^{\prime }$ area, focusing on the AF samples belonging exclusively to the auto-terms, as shown in Figure 2a for the signal $z_{\mathrm{SS}}$. Furthermore, the DBSCAN is able to successfully form $C_{\Omega }$ area with the additional AF samples outside the $N_{\tau }^{\prime }\times N_{\nu }^{\prime }$ area, emphasizing the non-LFM behavior of the signal $z_{\mathrm{RS}}$, as shown in Figure 2b. In addition, the AF samples that do not follow the auto-term trajectories in both examples have been classified as outliers/noise by the DBSCAN.

The comparisons between the TFDs obtained as a result of AF filtering with a full $N_{\tau }^{\prime }\times N_{\nu }^{\prime }$ rectangle (i.e., Γ=0,Υ=1) and the here-proposed CS-AF area selection with Γ and Υ parameters are shown in Figure 3. For the signal $z_{\mathrm{SS}}$, the $C_{\Omega }$ obtained with the here-proposed method reduced the number of artifacts while maintaining the auto-term resolution. On the other hand, the auto-term resolution and hyperbolic IF reconstruction in the signal $z_{\mathrm{RS}}$ have been significantly improved. Considering that the performance of the RTwIST algorithm is highly dependent on the distinction between the amplitude and bandwidth/duration of the auto-terms compared to the cross-terms, these improvements imply a better performance of the shrinkage operator in the detection of the samples related to the auto-terms, and, consequently, the overall reconstruction performance.

### 3.2. The Objective Function Performance

Here, the suitability of the objective functions is demonstrated for two cases: an oversparse and blurred reconstructed TFD, obtained with the RTwIST algorithm with arbitrarily chosen parameters. Note that the RTwIST algorithm was chosen because it showed superior performance to the other reconstruction algorithms in [24]. The discrepancy between the local number of components in the starting and reconstructed TFD, shown in Figure 4, indicates on an oversparse or blurred reconstructed TFD, as shown in Figure 5. More precisely, if the local number of components in the reconstructed TFD is less than in the starting TFD, the reconstructed TFD is oversparse with missing components. Moreover, the sharp drop in the local number of components to zero clearly indicates in which time or frequency slices the complete loss of components in the reconstructed TFD occurred, as shown in Figure 4a,b and Figure 5a. On the other hand, if the local number of components in the reconstructed TFD is greater than in the starting TFD, the reconstructed TFD contains artifacts and/or low-resolution auto-terms, as shown in Figure 4c,d and Figure 5b.

The numerical results for these two cases are given in Table 1, where the absence of the lower or the upper limit for the $M_{z}^{S}$ and *R*, or RN and HM measures, respectively, falsely classifies the oversparse TFD as high-performing. On the other hand, an increase in the ${\mathrm{MSE}}_{t}$, ${\mathrm{MSE}}_{f}$, and $N_{r}$ values solves this limitation and appropriately penalizes the oversparsity of the reconstructed TFD shown in Figure 5a. The low-resolution TFD with artifacts, which is usually correctly measured by $M_{z}^{S}$, *R*, RN, and HM, is also penalized by an increase in the value of ${\mathrm{MSE}}_{t}$, ${\mathrm{MSE}}_{f}$, and $N_{r}$. Additional performance indicators $({\mathrm{MAE}}_{t}$, ${\mathrm{MAE}}_{f}$, ${\mathrm{MAX}}_{t}$, and ${\mathrm{MAX}}_{f})$ also confirm the cases presented here.

### 3.3. Results

#### 3.3.1. Results for Synthetic Signal

The simulations have been performed for two cases: the CS-AF area with fixed geometry as proposed in [14] (Γ=0,Υ=1), and the CS-AF area selected by the here-proposed algorithm with optimized values of $\Gamma^{+}$ and $\Upsilon^{+}$ (${\left(\cdot \right)}^{+}$ denotes the optimized value). The reconstruction algorithms and CS-AF parameters have been optimized using the MOPSO algorithm resulting in the Pareto front from which the optimal solution was obtained using the FSM.

The numerical results of the comparison between the state-of-the-art reconstruction algorithms for the $z_{\mathrm{SS}}$ are given in Table 2 and shown in Figure 6. The RTwIST algorithm, with the optimized parameters and the proposed CS-AF area selection $\left(\alpha^{+},\beta^{+},p^{+},\delta_{t}^{+},\delta_{f}^{+},\Gamma^{+},\Upsilon^{+}\right)$ = (0.909,0.94,0.92,0.905,0.729,0.046,0.426), showed significant improvement compared to the same algorithm with the optimized parameters and the full $N_{\tau }^{\prime }\times N_{\nu }^{\prime }$ area (0.922, 0.881,0.94,0.882,0.9,0,1), especially for ${\mathrm{MSE}}_{t}$ and $N_{r}$, which are now reduced by 49.16% and 40.91%, respectively. The additional performance indicators confirm this improvement: ${\mathrm{MAE}}_{t}$ and ${\mathrm{MAE}}_{f}$ are reduced by 31.64% and 4.42%, respectively.

The superiority of the proposed CS-AF area is also evident in the remaining reconstruction algorithms. The SpaRSA algorithm achieves the most significant improvement, both numerically and visually. Namely, ${\mathrm{MSE}}_{t}$, ${\mathrm{MSE}}_{f}$, and $N_{r}$ values are reduced by 70.45%, 59.38%, and 98.39%, respectively. Furthermore, the importance of $N_{r}$ is evident in the results obtained for the SALSA algorithm. Although the results show a slight increase in the ${\mathrm{MSE}}_{f}$ value (by 5.29%), which is a consequence of the lower auto-term resolution, a significant reduction in $N_{r}$ is achieved (by 93.33%), indicating higher component consistency with fewer interference and noise samples. Moreover, the SpaRSA and TwIST algorithms with the proposed CS-AF area resulted in the smallest $N_{r}$ values, which almost correspond to the ideal global number of components. Finally, the RTwIST algorithm achieved the best overall performance among the tested algorithms. Upon conducting a visual inspection of the outcomes presented in Figure 6, the aforementioned findings were further substantiated. Specifically, it was observed that by disregarding the AF samples that were associated with interference or noise in the obtained CS-AF areas, they were prevented from resurfacing in the obtained reconstructed TFDs. Additionally, this approach resulted in either the retention of or a slight enhancement in the resolution and preservation of the auto-terms.

Moreover, the increase in $M_{z}^{S}$ for the SALSA algorithm with the proposed CS-AF area shows the limitation of the global concentration approach. This limitation is also apparent when comparing the results for the optimized RTwIST with the oversparse RTwIST from Table 1—all global measures erroneously favor the oversparse reconstructed TFD.

Note that a significant reduction in the CS-AF area size has been achieved, from 37.74% for the SpaRSA algorithm up to 79.55% for the SALSA algorithm, proving that most of the AF samples within the full $N_{\tau }^{\prime }\times N_{\nu }^{\prime }$ rectangle are not related to the auto-terms. Furthermore, the proposed CS-AF area selection with given $\Gamma^{+}$ and $\Upsilon^{+}$ values did not increase the execution time of reconstruction algorithms. In fact, competitive reconstruction execution time has been achieved for all algorithms when using the proposed CS-AF, with the SpaRSA being the fastest reconstruction algorithm.

#### 3.3.2. Results for Real-Life Signal

Next, the same simulations have been performed on the real-life gravitational-wave signal $z_{\mathrm{RS}}$ [56,57]. The optimal solution for the RTwIST algorithm obtained by the FSM gives: $\alpha^{+}=0.946$, $\beta^{+}=0.92$, $p^{+}=1$, $\delta_{t}^{+}=0.91$, $\delta_{f}^{+}=0.56$, $\Gamma^{+}=0.142$, $\Upsilon^{+}=0.189$. The numerical results of the comparison are provided in Table 3. All reconstruction algorithms with the proposed CS-AF area achieved reductions in ${\mathrm{MSE}}_{t}$ and ${\mathrm{MSE}}_{f}$ values by up to 90.41% and 73.95% for the RTwIST algorithm, respectively, with little or no reduction in $N_{r}$ (except for the RTwIST algorithm, which achieved higher $N_{r}$). The results follow the observations from the previous example; the RTwIST algorithm has the best performance, while the SpaRSA algorithm has the smallest execution time. By conducting a visual examination of the results illustrated in Figure 7, it is evident that utilizing the full $N_{\tau }^{\prime }\times N_{\nu }^{\prime }$ rectangle results in considerable degradation of the hyperbolic IF demonstrated in Figure 3d. This degradation is also observed in all reconstructed TFDs. However, upon implementing the proposed CS-AF area selection approach, a substantial enhancement in the component concentration along the hyperbolic IF is observed in all the reconstructed TFDs.

In addition, all reconstruction algorithms achieved better results with up to 16% fewer CS-AF samples. The reduction in the CS-AF area is less significant than in the previous example. This is because the optimized CS-AF area in this example includes additional auto-term-related samples outside the $N_{\tau }^{\prime }\times N_{\nu }^{\prime }$ rectangle, which improves the hyperbolic IF reconstruction of the auto-term.

The SpaRSA algorithm with the proposed CS-AF area and SALSA with both CS-AF areas have almost equal *R* values. This result confirms two properties of the global Rényi entropy [31]: its insensitivity to small changes in the TFDs and permutation symmetry (the permutation of any probability in the distribution does not affect the entropy value). Again, the global concentration measures incorrectly highlighted the SALSA with (Γ,Υ)=(0,1) and incomplete auto-term as better performing than the SALSA with $\left(\Gamma^{+},\Upsilon^{+}\right)=\left(0.098,0.195\right)$. This further emphasizes the importance of the information about the auto-terms obtained from the localized approach (${\mathrm{MSE}}_{t}$, ${\mathrm{MSE}}_{f}$, and $N_{r}$) over the global approach in evaluating similar TFDs.

Finally, Table 4 reports the results obtained for the $z_{\mathrm{SS}}$ embedded in AWGN with three different SNRs. The lowest SNR tested here was chosen to be 2 dB, as the investigation in [24] showed that the performance of the RTwIST algorithm is reliable for SNR >1 dB. The results show that when the SNR decreases, the ${\mathrm{MSE}}_{t}$, ${\mathrm{MSE}}_{f}$, and $N_{r}$ values increase for all CS-AF selections. However, the improvement of the proposed CS-AF area selection over the full $N_{\tau }^{\prime }\times N_{\nu }^{\prime }$ area is largest for the lowest SNR =2 dB—${\mathrm{MSE}}_{t}$, ${\mathrm{MSE}}_{f}$, and $N_{r}$ values are reduced by 45.63%, 34.17%, and 43.24%, respectively. The reason is that by increasing the noise level, more noise-related samples are present in the $N_{\tau }^{\prime }\times N_{\nu }^{\prime }$, which are then excluded using the proposed CS-AF area selection. Note also that the optimal parameter value $\Gamma^{+}$ increases with decreasing SNR. This indicates that the noise-related samples are of higher magnitude and therefore need to be excluded with a higher Γ value.

### 3.4. Results Summary

The proposed CS-AF area selection approach outperforms the existing $N_{\tau }^{\prime }\times N_{\nu }^{\prime }$ rectangle [14] in terms of visual inspection and proposed measures ${\mathrm{MSE}}_{t}$, ${\mathrm{MSE}}_{f}$, and $N_{r}$, as visible from Figure 6 and Figure 7 and corroborated by numerical results in Table 2 and Table 3. For both signals considered, the DBSCAN algorithm successfully extracted a single cluster that satisfied the key properties of the auto-terms in the AF, and the obtained CS-AF areas showed to be capable of tracking the trajectories of the auto-terms, allowing an arbitrary shape of the area.

For the signal $z_{\mathrm{SS}}$, the reconstruction improvement was achieved by excluding the AF samples not associated with auto-terms from the full $N_{\tau }^{\prime }\times N_{\nu }^{\prime }$ rectangle. This attenuated or slightly improved the auto-terms resolution, while significantly reducing the reconstruction of samples related to interference and noise, as measured with ${\mathrm{MSE}}_{t}$, ${\mathrm{MSE}}_{f}$, and $N_{r}$ which were improved by up to 98.39%. In addition, the CS-AF area was reduced by up to 79.55%, proving that the existing $N_{\tau }^{\prime }\times N_{\nu }^{\prime }$ rectangle may include a significant amount of samples not related to the auto-terms.

For the real-life signal $z_{\mathrm{RS}}$, the reconstruction improvement was achieved by including auto-term-related AF samples outside the $N_{\tau }^{\prime }\times N_{\nu }^{\prime }$ area, which turned out to be essential for highly non-LFM components. The proposed CS-AF area selection approach overcomes the limitations of the existing $N_{\tau }^{\prime }\times N_{\nu }^{\prime }$ rectangle, which adapts only around the linear part of the auto-term behavior in the AF, and presents a strong basis for future investigation of gravitational-wave signals with deep learning methods using time-frequency analysis, such as in [52].

Experimental results with several state-of-the-art reconstruction algorithms, namely, the RTwIST [24], TwIST [20], SpaRSA [53], and SALSA [54], have shown that the reconstruction improvement by the proposed CS-AF area selection is not limited to a single reconstruction algorithm. In particular, the RTwIST algorithm exhibited the best performance among the considered algorithms, achieving the highest objective function values, while the SpaRSA algorithm achieved the fastest execution time. This finding is consistent with previous studies that have shown the superior performance of the RTwIST algorithm in signal reconstruction tasks [24,28].

The tests conducted for different SNR levels showed consistent improvement in the proposed CS-AF area selection, which is more significant for lower SNR values as more noise samples are present in the AF.

The optimization algorithm’s time primarily depends on the execution time of the reconstruction algorithm, the number of objectives, the number of particles, and the number of iterations. In this paper, the optimization was conducted offline to find optimal parameters for high-performing reconstructed TFDs which might not be identified through manual selection. The average time for the entire optimization process with parameters defined at the beginning of this section ranged from 4042 s for the SpaRSA algorithm and signal $z_{\mathrm{RS}}$ to 9711 s for the RTwIST algorithm and signal $z_{\mathrm{SS}}$.

Furthermore, we showed that the ${\mathrm{MSE}}_{t}$ and ${\mathrm{MSE}}_{f}$ measures, in addition to preserving the component, provide information about their resolution and cross-term suppression quality as well. TFD component connectivity was ensured by measuring the number of regions with continuously connected samples $N_{r}$. These measures overcome the limitations of global concentration measures and are suitable as objective functions for a multi-objective optimization algorithm. However, it is important to note that the interpretation of global measures should always be supported by theoretical proof that the utilized approach does not involve the absence of auto-terms, or by an additional measure proving that the auto-terms have been preserved.

## 4. Conclusions

This paper presents the CS-AF area construction with adaptive geometry, which selects only the most impactful AF samples related to the auto-terms. The selected AF samples’ key properties have been ensured using the DBSCAN algorithm. The proposed approach has improved the TFD reconstruction for all considered algorithms while using significantly fewer CS-AF samples compared to the geometrically fixed CS-AF area.

These findings have been supported by the introduced TFD performance measures. Namely, a localized approach of measuring TFD concentration has been used through the local number of components obtained from the STRE and NBRE. Low MSE values obtained for the local number of components in starting and reconstructed TFDs indicate higher resolution and better component consistency in the reconstructed TFD, which precludes the need for a global concentration measure. Moreover, the TFD components’ connectivity has been enforced by measuring the number of regions with continuously connected samples. The obtained results have shown that these criteria provide suitable objective functions for multi-objective meta-heuristic optimization in sparse TFD reconstruction and overcome the limitations of global concentration measures.

The presented results show that the optimization method with the proposed objective functions and the CS-AF area selection achieve highly concentrated TFDs with good auto-term preservation and consistency for both noisy synthetic and real-life signals, without requiring prior knowledge of the signal nature. The multi-objective optimization framework presented in this study is not limited to sparse TFD reconstruction. Hence, future work will focus on applying our approach to other TF methods and developing a single measure for auto-term loss evaluation, enabling less-demanding single-objective optimization.

## Figures and Tables

**Figure 1 sensors-23-04148-f001:**
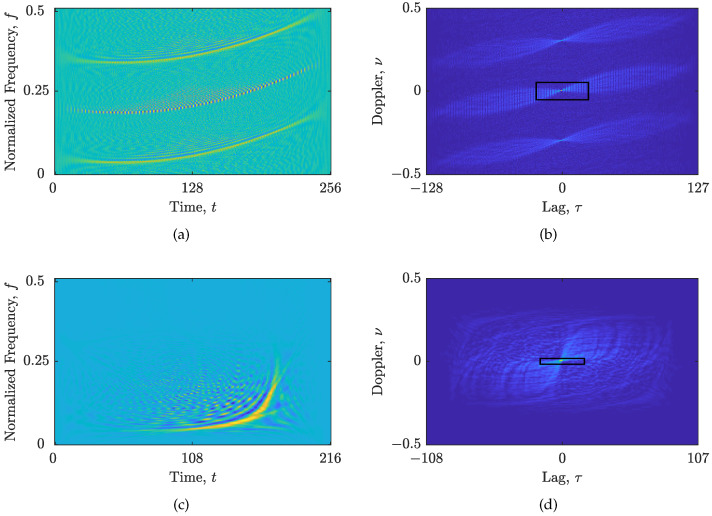
Considered signals: (**a**) WVD of $z_{\mathrm{SS}}$; (**b**) AF of $z_{\mathrm{SS}}$, $N_{\tau }^{\prime }=49$, $N_{\nu }^{\prime }=53$; (**c**) WVD of $z_{\mathrm{RS}}$; (**d**) AF of $z_{\mathrm{RS}}$, $N_{\tau }^{\prime }=35$, $N_{\nu }^{\prime }=15$. The automatically selected $N_{\tau }^{\prime }\times N_{\nu }^{\prime }$ AF area is marked by a rectangle.

**Figure 2 sensors-23-04148-f002:**
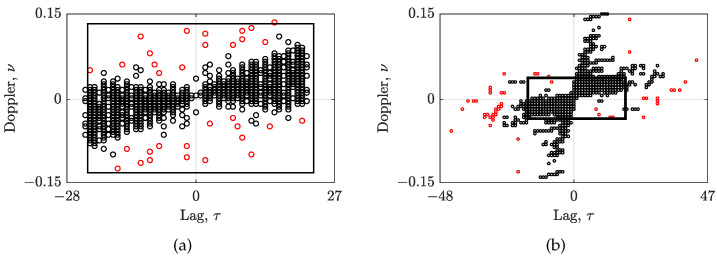
The clustered CS-AF area $C_{\Omega }$, whose samples are marked by black circles, of: (**a**) $z_{\mathrm{SS}}$, with Γ=0.06,Υ=0.5; (**b**) $z_{\mathrm{RS}}$ with Γ=0.15,Υ=0.18. Outlier/noise samples are marked by red circles, while the automatically selected $N_{\tau }^{\prime }\times N_{\nu }^{\prime }$ area is marked by a black rectangle.

**Figure 3 sensors-23-04148-f003:**
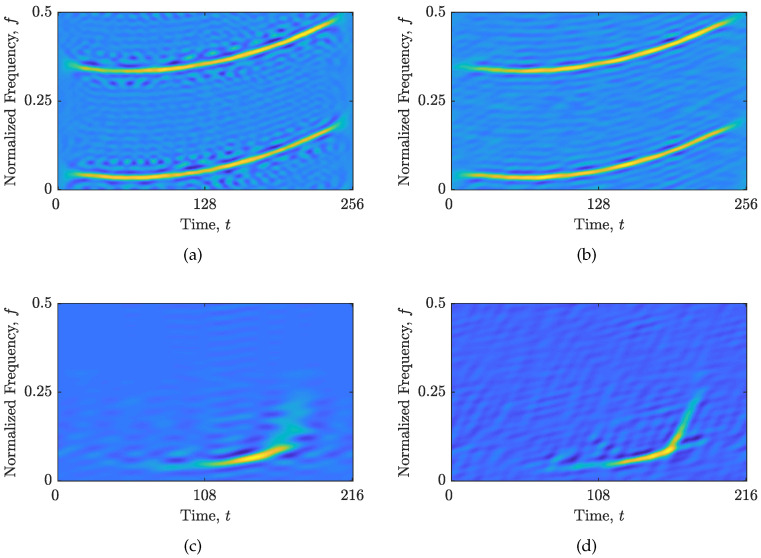
The obtained TFD from the filtered AF of: (**a**) the $z_{\mathrm{SS}}$ with a full $N_{\tau }^{\prime }\times N_{\nu }^{\prime }$ area, Γ=0,Υ=1; (**b**) the $z_{\mathrm{SS}}$ with the proposed CS-AF area, Γ=0.06,Υ=0.5; (**c**) the $z_{\mathrm{RS}}$ with a full $N_{\tau }^{\prime }\times N_{\nu }^{\prime }$ area, Γ=0,Υ=1; (**d**) the $z_{\mathrm{RS}}$ with the proposed CS-AF area, Γ=0.15,Υ=0.18.

**Figure 4 sensors-23-04148-f004:**
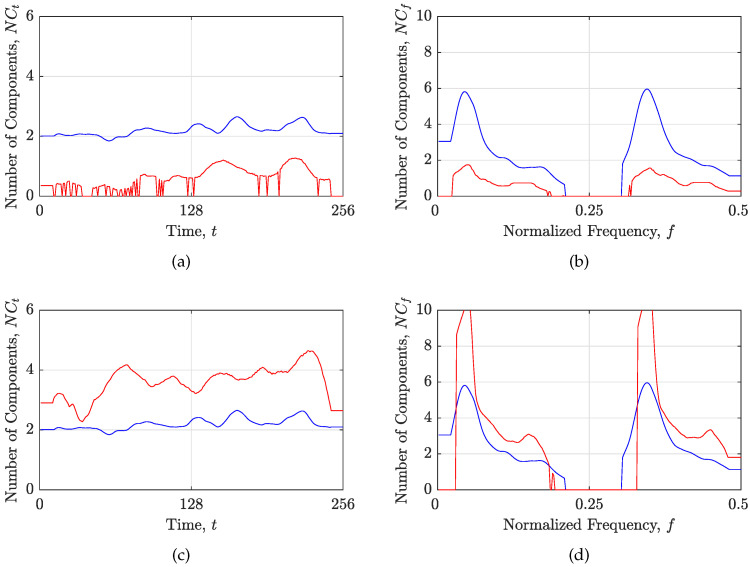
The local number of components in the starting (blue line) and the reconstructed (red line) TFD based on the RTwIST algorithm for the signal $z_{\mathrm{SS}}$, with α=0.84,β=0.69 and: (**a**,**b**) $p=0.75,\delta_{t}=0.99,\delta_{f}=0.99,\Gamma =0.25,\Upsilon =1$; (**c**,**d**) $p=0.1,\delta_{t}=0.75,\delta_{f}=0.75,\Gamma =0.1,\Upsilon =1$.

**Figure 5 sensors-23-04148-f005:**
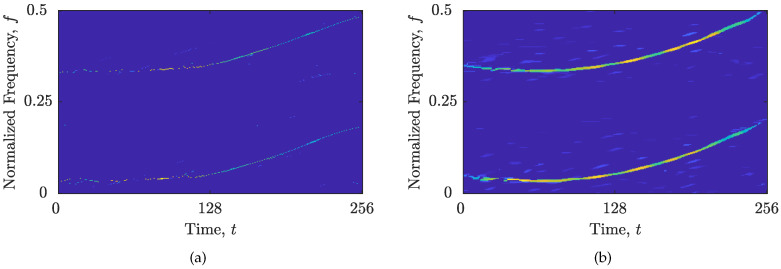
Reconstructed sparse TFDs based on the RTwIST algorithm for the considered test signal $z_{\mathrm{SS}}$, with α=0.84,β=0.69 and: (**a**) $p=0.75,\delta_{t}=0.99,\delta_{f}=0.99,\Gamma =0.25,\Upsilon =1$; (**b**) p=0.1,$\delta_{t}=0.75,\delta_{f}=0.75,\Gamma =0.1,\Upsilon =1$.

**Figure 6 sensors-23-04148-f006:**
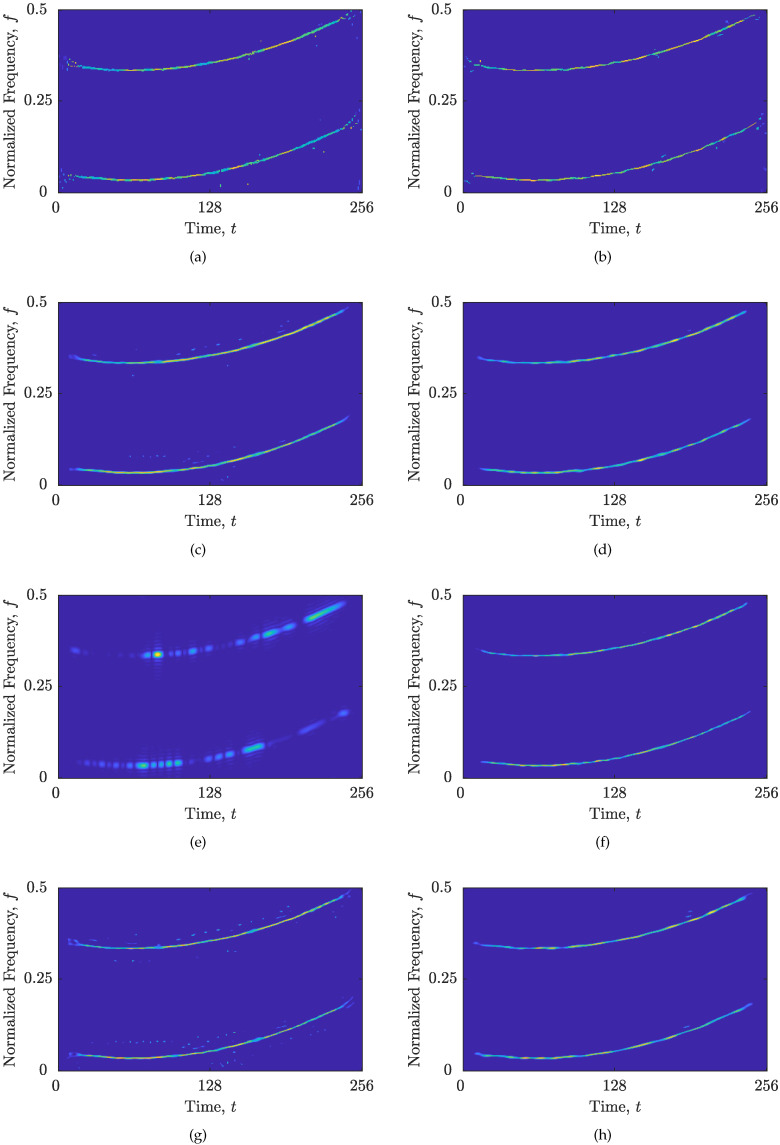
Reconstructed sparse TFDs of $z_{\mathrm{SS}}$ with: (**a**) RTwIST algorithm, Γ=0,Υ=1; (**b**) RTwIST algorithm, $\Gamma^{+}=0.046,\Upsilon^{+}=0.426$; (**c**) TwIST algorithm, Γ=0,Υ=1; (**d**) TwIST algorithm, $\Gamma^{+}\;=\;0.043,$$\Upsilon^{+}=0.520$; (**e**) SpaRSA algorithm, Γ=0,Υ=1; (**f**) SpaRSA algorithm, $\Gamma^{+}\;=\;0.029,\Upsilon^{+}=0.515$; (**g**) SALSA algorithm, Γ=0,Υ=1; (**h**) SALSA algorithm, $\Gamma^{+}=0.070,\Upsilon^{+}=0.350$.

**Figure 7 sensors-23-04148-f007:**
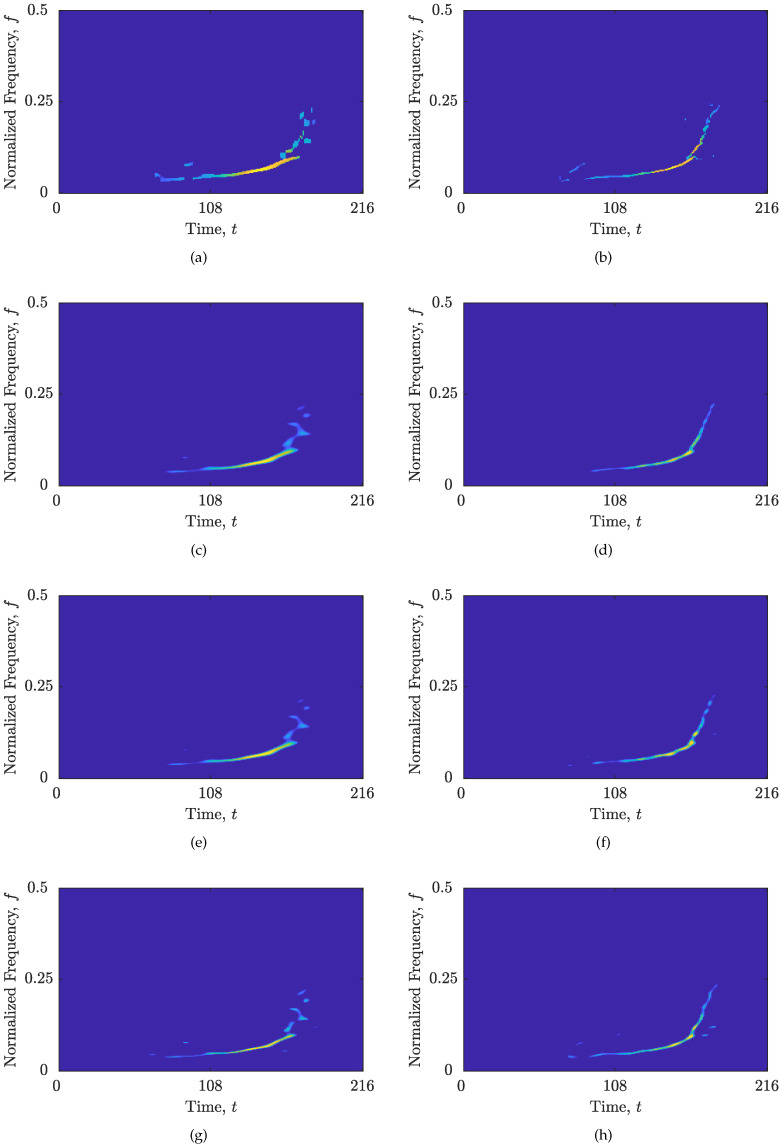
Reconstructed sparse TFDs of $z_{\mathrm{RS}}$ with: (**a**) RTwIST algorithm, Γ=0,Υ=1; (**b**) RTwIST algorithm, Γ=0.142,Υ=0.189; (**c**) TwIST algorithm, Γ=0,Υ=1; (**d**) TwIST algorithm, Γ=0.3,Υ=0.179; (**e**) SpaRSA algorithm, Γ=0,Υ=1; (**f**) SpaRSA algorithm, Γ=0.08,Υ=0.195; (**g**) SALSA algorithm, Γ=0,Υ=1; (**h**) SALSA algorithm, Γ=0.098,Υ=0.195.

**Table 1 sensors-23-04148-t001:** Comparison of the oversparse and low-resolution TFDs.

Signal $z_{\mathrm{SS}}$		
RTwIST, α=0.84,β=0.69		
	p=0.75	p=0.01
	$\delta_{t}=0.99,\delta_{f}=0.99$	$\delta_{t}=0.75,\delta_{f}=0.75$
	Γ=0.25,Υ=1	Γ=0.1,Υ=1
$\mathrm{card}(C_{\Omega })$	39	397
${\mathrm{MSE}}_{t}$	0.3784	0.1070
${\mathrm{MSE}}_{f}$	0.1178	0.0301
$N_{r}$	168	103
${\mathrm{MAE}}_{t}$	0.6061	0.3117
${\mathrm{MAE}}_{f}$	0.2649	0.1204
${\mathrm{MAX}}_{t}$	0.8965	0.5133
${\mathrm{MAX}}_{f}$	0.7524	0.4832
*R*	8.299	10.938
$M_{z}^{S}$	0.0075	0.0459
$RN(\times 10^{3})$	4.10	0.65
HM	0.9321	0.8239

**Table 2 sensors-23-04148-t002:** Performance of the state-of-the-art reconstruction algorithms with the full $N_{\tau }^{\prime }\times N_{\nu }^{\prime }$ area ((Γ,Υ)=(0,1)) versus the proposed CS-AF area selection $\left(\Gamma^{+},\Upsilon^{+}\right)$.

Signal $z_{\mathrm{SS}}$								
	**RTwIST [24]**		**TwIST [20]**		**SpaRSA [53]**		**SALSA [54]**	
	Γ=0	$\Gamma^{+}=0.046$	Γ=0	$\Gamma^{+}=0.043$	Γ=0	$\Gamma^{+}=0.029$	Γ=0	$\Gamma^{+}=0.070$
	Υ=1	$\Upsilon^{+}=0.426$	Υ=1	$\Upsilon^{+}=0.520$	Υ=1	$\Upsilon^{+}=0.515$	Υ=1	$\Upsilon^{+}=0.350$
$\mathrm{card}(C_{\Omega })$	2597	883	2597	1009	2597	1617	2597	531
${\mathrm{MSE}}_{t}$	0.0179	**0.0091**	0.0597	0.0378	0.1029	0.0253	0.0159	0.0117
${\mathrm{MSE}}_{f}$	0.0178	**0.0158**	0.0207	0.0187	0.0904	0.0248	0.0189	0.0199
$N_{r}$	22	13	35	**3**	186	**3**	75	5
${\mathrm{MAE}}_{t}$	0.1141	**0.078**	0.2344	0.1738	0.2866	0.0847	0.1054	0.0928
${\mathrm{MAE}}_{f}$	0.0769	**0.0735**	0.0886	0.0792	0.224	0.091	0.0742	0.0778
${\mathrm{MAX}}_{t}$	0.3192	**0.2654**	0.5791	0.6475	0.5833	0.789	0.6804	0.2983
${\mathrm{MAX}}_{f}$	0.5334	0.6170	**0.4897**	0.5411	0.9411	0.6762	0.682	0.6112
*R*	10.33	9.97	10.47	10.28	11.44	**9.94**	10.22	10.14
$M_{z}^{S}$	0.0269	0.0195	0.0278	0.0237	0.369	**0.0185**	0.0405	0.0431
$RN(\times 10^{3})$	0.89	1.13	0.92	1.21	0.03	**1.30**	1.11	1.22
HM	0.8607	**0.8781**	0.8516	0.8611	0.6578	0.8771	0.8631	0.8672
t[s] *	0.747	0.726	0.231	0.233	0.465	**0.117**	0.467	0.385

* In order to minimize the stochastic behavior of the reconstruction algorithm execution times, the results have been averaged over 1000 algorithm runs. The simulations were performed on a PC with the AMD Ryzen 7 3700X @ 3.60 GHz processor and 16 GB of RAM. Values in bold indicate the best-performing or the fastest algorithm.

**Table 3 sensors-23-04148-t003:** Performance of the state-of-the-art reconstruction algorithms with the full $N_{\tau }^{\prime }\times N_{\nu }^{\prime }$ area ((Γ,Υ)=(0,1)) versus the proposed CS-AF area selection $\left(\Gamma^{+},\Upsilon^{+}\right)$.

Signal $z_{\mathrm{RS}}$								
	**RTwIST [24]**		**TwIST [20]**		**SpaRSA [53]**		**SALSA [54]**	
	Γ=0	$\Gamma^{+}=0.142$	Γ=0	$\Gamma^{+}=0.130$	Γ=0	$\Gamma^{+}=0.080$	Γ=0	$\Gamma^{+}=0.098$
	Υ=1	$\Upsilon^{+}=0.189$	Υ=1	$\Upsilon^{+}=0.179$	Υ=1	$\Upsilon^{+}=0.195$	Υ=1	$\Upsilon^{+}=0.195$
$\mathrm{card}(C_{\Omega })$	525	441	525	505	525	505	525	456
${\mathrm{MSE}}_{t}$	0.0386	**0.0037**	0.0404	0.0268	0.0356	0.0286	0.0155	0.0150
${\mathrm{MSE}}_{f}$	0.0357	**0.0093**	0.028	0.0200	0.0245	0.0197	0.0209	0.0144
$N_{r}$	9	10	4	**2**	4	4	7	7
${\mathrm{MAE}}_{t}$	0.1375	**0.0332**	0.1221	0.1084	0.1192	0.1128	0.0733	0.0773
${\mathrm{MAE}}_{f}$	0.1119	**0.0510**	0.1002	0.0820	0.0952	0.0807	0.0845	0.0627
${\mathrm{MAX}}_{t}$	0.5148	**0.2061**	0.5371	0.5191	0.5212	0.553	0.4898	0.451
${\mathrm{MAX}}_{f}$	0.5566	**0.5092**	0.5538	0.6227	0.5130	0.5747	0.6683	0.6087
*R*	8.82	**8.01**	8.6	8.03	8.38	8.12	8.14	8.13
$M_{z}^{S}$	0.0134	**0.0070**	0.0130	0.0078	0.0109	0.0085	0.0344	0.0473
$RN(\times 10^{3})$	2.91	5.22	4.12	5.53	4.62	5.23	**5.74**	5.61
HM	0.9024	**0.9272**	0.9075	0.9261	0.9150	0.9232	0.9230	0.9221
t[s] *	0.203	0.331	0.133	0.250	0.060	**0.050**	0.162	0.137

* In order to minimize the stochastic behavior of the reconstruction algorithm execution times, the results have been averaged over 1000 algorithm runs. The simulations were performed on a PC with the AMD Ryzen 7 3700X @ 3.60 GHz processor and 16 GB of RAM. Values in bold indicate the best-performing or the fastest algorithm.

**Table 4 sensors-23-04148-t004:** Performance comparison of the RTwIST algorithm with the full $N_{\tau }^{\prime }\times N_{\nu }^{\prime }$ area ((Γ,Υ)=(0,1)) versus the proposed CS-AF area selection $\left(\Gamma^{+},\Upsilon^{+}\right)$ for the $z_{\mathrm{SS}}$ embedded in noise with different SNRs.

Signal $z_{\mathrm{SS}}$, RTwIST						
	**2 dB**		**6 dB**		**10 dB**	
	Γ=0	$\Gamma^{+}=0.071$	Γ=0	$\Gamma^{+}=0.044$	Γ=0	$\Gamma^{+}=0.036$
	Υ=1	$\Upsilon^{+}=0.336$	Υ=1	$\Upsilon^{+}=0.452$	Υ=1	$\Upsilon^{+}=0.454$
${\mathrm{MSE}}_{t}$	0.0320	0.0174	0.0152	0.0104	0.0102	0.0081
${\mathrm{MSE}}_{f}$	0.0321	0.0211	0.0188	0.0161	0.0131	0.0112
$N_{r}$	37	21	20	12	15	9

Values are averaged from 1000 independent algorithm runs of the signal with different noise realizations.

## Data Availability

Not applicable.

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
