# Peer review of "Sparse Time-Frequency Distribution Reconstruction Using the Adaptive Compressed Sensed Area Optimized with the Multi-Objective Approach"

_sensors, 2023, doi:10.3390/s23084148_

Round 1

Reviewer 1 Report

Dear Authors,

I have some comments on your article:

1. All indexes in symbols in text and equations should be checked carefully.

2. Literature should be checked if there are no newer items. Especially from the last 18 months.

3. I miss the discussion of the results and the summary of a real example showing the usefulness of the proposed method.

Reviewer 2 Report

The abstract is added with a brief research method, brief research results and brief conclusions

Introduction needs to be further strengthened by adding the benefits of this research, its use in industry and research objectives related to scientific development

Figure 1 needs to be enlarged to make it clearer

Figure 3 needs to be enlarged to make it clearer

Figure 5 needs to be enlarged to make it clearer

Figure 6 needs to be enlarged to make it clearer

Section 3.4 Results Summary, it is necessary to add cited references

Conclusions can be made shorter and general in nature, answering research questions

Reviewer 3 Report

This paper used  a density-based spatial clustering algorithm to select compressive sensing signal ambiguity function region, and was written in detail, but there are some problems:
The description should be more concise and objective, and the word "we" should not appear.
The full name of DBSCAN in line 45 is not correct.

Reviewer 4 Report

This study proposed a CS-AF area construction approach for an adaptive geometry of AF samples. The key properties of the selected impactful samples were ensured through the DBSCAN algorithm. Examples demonstrated that the proposed approach have improved the TED reconstruction for significantly fewer CS-AF samples, thus largely enhancing the reconstruction precision with fewer sample data. This study is meaningful and the approach is of some novelty. Generally, this manuscript is well written. My particular comments are as follows:

1.     In the reconstruction algorithm, the treatment balance between cross-terms and auto-term resolution is important. More discussions are expected in the examples.

2.     How to determine the time-localization operator and frequency-localization operator and other thresholding values in the examples?

3.     Some details on the multi-objective meta-heuristic optimization to get the Pareto front are expected and time spent in this optimization should be discussed.

Reviewer 5 Report

The manuscript presents an important research problem. 

The authors correctly discussed the research problem and extensively presented the mathematical foundations and included many interesting research results.

The manuscript as presented is printable.

Best Regards
